# Shear-margin melting causes stronger transient ice discharge than ice-stream melting in idealized simulations

Johannes Feldmann[1], Ronja Reese[1,2], Ricarda Winkelmann[1,3], and Anders Levermann[1,3,4]

[1]Potsdam Institute for Climate Impact Research (PIK), Potsdam, Germany
[2]Department of Geography and Environmental Sciences, Northumbria University, Newcastle, UK
[3]Institute of Physics, University of Potsdam, Potsdam, Germany
[4]LDEO, Columbia University, New York, USA

*Correspondence to:* Johannes Feldmann (johannes.feldmann@pik-potsdam.de)

**Abstract.** Basal ice-shelf melting is the key driver of Antarctica's increasing sea-level contribution. In diminishing the buttressing force of the ice shelves that fringe the ice sheet, the melting increases the ice discharge into the ocean. Here we contrast the influence of basal melting in two different ice-shelf regions on the time-dependent response of an isothermal, inherently buttressed ice-sheet-shelf system. In the idealized numerical simulations, the basal-melt perturbations are applied close to the grounding line in the ice-shelf's 1) ice-stream region, where the ice shelf is fed by the fastest ice masses that stream through the upstream bed trough and 2) shear margins, where the ice flow is slower. The results show that melting below one or both of the shear margins can cause a decadal to centennial increase in ice discharge that is more than twice as large compared to a similar perturbation in the ice-stream region. We attribute this to the fact that melt-induced ice-shelf thinning in the central grounding-line region is attenuated very effectively by the fast flow of the central ice stream. In contrast, the much slower ice dynamics in the lateral shear margins of the ice shelf facilitate sustained ice-shelf thinning and thereby foster buttressing reduction. Regardless of the melt location, a higher melt concentration toward the grounding line generally goes along with a stronger response. Our results highlight the vulnerability of outlet glaciers to basal melting in stagnant, buttressing-relevant ice-shelf regions, a mechanism that may gain importance under future global warming.

## 1 Introduction

Virtually all of Antarctica's observed sea-level contribution comes from increased discharge of ice into the ocean (Rignot et al., 2019; The IMBIE Team, 2020). The discharge is regulated by the floating ice shelves that fringe the ice sheet and exert a buttressing force on the upstream outlet glaciers that drain the ice sheet (Dupont and Alley, 2005; Schoof, 2007; Goldberg et al., 2009; Favier et al., 2012; Gudmundsson et al., 2012; Haseloff and Sergienko, 2018; Pegler, 2018; Reese et al., 2018). Increased basal melting and thus thinning of buttressing ice shelves (Rignot et al., 2013; Paolo et al., 2015) reduces their backforce which can lead to speed-up, thinning and retreat of the upstream grounded masses (Shepherd et al., 2002; Jenkins et al., 2010; Joughin and Alley, 2011; Rignot et al., 2014; Konrad et al., 2018; Gudmundsson et al., 2019).

Future atmospheric warming will likely increase the oceanic heat content available for sub-ice-shelf melting (Rignot and Jacobs, 2002; Hellmer et al., 2012; Spence et al., 2014; Schmidtko et al., 2014; Naughten et al., 2018). Increased melting may

lead to increased ice discharge and thus contribute positively to future sea level rise (e.g., Bindschadler et al., 2013; Bamber and Aspinall, 2013; Joughin et al., 2014; Favier et al., 2014; Mengel and Levermann, 2014; Pollard et al., 2015; Bakker et al., 2017; Jackson et al., 2018; Levermann et al., 2020; Seroussi et al., 2020; Payne et al., 2021; Edwards et al., 2021).

Observations of Antarctic outlet glaciers show that sub-ice-shelf melt rates are typically strongest close to the grounding line where fast and thick ice masses cross the outlets' central grounding-line section (Dutrieux et al., 2013; Shean et al., 2019), which might imply that also melt rate changes and thinning are strongest in those areas. However, ice-shelf melting as well as thinning patterns can be spatially very heterogeneous (Pritchard et al., 2012; Dutrieux et al., 2013; Paolo et al., 2015), including the possibility of comparatively strong melting in the more stagnant parts and the shear zones of an ice shelf, e.g., regions at the lateral margins of the fast-flowing ice streams or regions close to ice rises (Berger et al., 2017; Goldberg et al., 2019; Shean et al., 2019; Alley et al., 2019; Adusumilli et al., 2020). Observations show that, especially in West Antarctica, ice-shelf shear margins are favorable locations for the formation of basal channels in which warm-water flow leads to enhanced melting (Sergienko, 2013; Alley et al., 2016).

A number of numerical studies demonstrated that the location and the distribution of ice-shelf thinning have strong influence on the reduction of the ice shelf's buttressing strength and the corresponding ice-sheet response. For instance, Gagliardini et al. (2010) found in conceptual flowline simulations that the grounding-line position and the volume of an ice sheet are sensitive to changes in the degree of concentration of the melting to the grounding line even if the average melt magnitude remains the same. Reese et al. (2018) conducted diagnostic perturbation experiments to assess the instantaneous response of the integrated flux across the grounding line of the Antarctic Ice Sheet to local melt perturbations of its ice shelves. Their results indicate that in general perturbations closer to the grounding lines induced stronger responses. Strongest flux responses were induced in regions close to the grounding lines of ice streams but high responses were also found in regions close to pinning points or shear margins. Zhang et al. (2020) used the same perturbation method and compared it to an adjoint-based approach which allows for higher spatial resolution. They applied both approaches in an idealized setup and a real-world setup of Larsen C, showing their consistency and finding also that the integrated grounding-line flux is most sensitive to ice-shelf thinning close to the grounding line. Using the adjoint-based method in short-term prognostic simulations of Crosson and Dotson ice shelves and their feeding glaciers in West Antarctica, Goldberg et al. (2019) found that the linearized response of the glaciers' sea-level relevant ice volume over 15 years is, consistently with the instantaneous studies by Reese et al. (2018) and Zhang et al. (2020), most sensitive to ice-shelf melting close to the grounding lines and regions of high horizontal shearing. While these studies clearly suggest that the effect of ice-shelf thinning on ice-shelf buttressing matters most for specific thinning locations (i.e., close to the grounding line and in shear zones) their results are limited to the instantaneous (short-term) response.

Here we carry out transient simulations to investigate the longer-term (100-yr) response. That is, we model an inherently buttressed, idealized ice-sheet-shelf system to compare the effects of basal melting in the central ice-stream region vs. the lateral shear-margin regions within an ice-shelf embayment that buttresses an upstream outlet glacier. While the adjoint method and the diagnostic studies mentioned above provide diagnostic, instantaneous ice-flux sensitivities to melt perturbations, our method allows to model the response of the ice-sheet-shelf system over time, i.e., the transient response of the ice geometry, buttressing, ice flux and grounding-line position. Besides altering the melt location (beneath the ice stream / beneath one

or both shear margins) we also vary the magnitude and the spatial extent of the perturbation. The numerical model and the experimental design are outlined in Sec. 2. The results are analyzed in Sec. 3 and discussed in Sec. 4 where we also draw our conclusions.

## 2 Methods

### 2.1 Numerical model

We use the open-source Parallel Ice Sheet Model (PISM; Bueler and Brown, 2009; Winkelmann et al., 2011; Khroulev and Authors, 2020), version stable1.0 (https://github.com/pism/pism/). The model applies a superposition of the shallow-ice approximation (SIA; Morland, 1987) and the shallow-shelf approximation (SSA; Hutter, 1983) of the Stokes stress balance (Greve and Blatter, 2009). In particular, the SSA allows for stress transmission across the grounding line and thus accounts for the buttressing effect of laterally confined ice shelves on the upstream grounded regions (Gudmundsson et al., 2012; Fürst et al., 2016; Reese et al., 2018). The model applies a linear interpolation of the freely evolving grounding line and accordingly interpolated basal friction (Feldmann et al., 2014). Grounding-line migration has been evaluated in the model intercomparison exercises MISMIP3d (Pattyn et al., 2013; Feldmann et al., 2014) and MISMIP+ (Asay-Davis et al., 2016; Cornford et al., 2020). To improve the approximation of driving stress across the grounding line, the surface gradient is calculated using centered differences of the ice thickness across the grounding line (Reese et al., 2020).

### 2.2 Setup and experimental design

The model is initiated with a domain-wide layer of ice of $500 \, \mathrm{m}$ thickness from which the ice-sheet-shelf system evolves, reaching equilibrium after several thousand model years. The prescribed surface mass balance and ice softness are constant in space and time (see Table 1 for more parameters). Basal friction is calculated according to a Weertman-type power law (Asay-Davis et al., 2016, Eq. 6). The prescribed bed topography is taken from MISMIP+ (Asay-Davis et al., 2016, Eq. 1) which is a smaller (half as long and narrower) version of the one from Gudmundsson et al. (2012). It is designed to model an idealized, strongly buttressed, marine ice-sheet-shelf system (Fig. 1). The grounded ice sheet is drained by an ice stream (Fig. 2) through a bed trough, feeding a bay-shaped ice shelf which calves into the ocean. The bed topography is a superposition of two components: the bed elevation in $x$-direction is overall declining from the ice divide towards the ocean but has an overdeepening (landward down-sloping bed section) just upstream of the continental shelf break. The bed component in $y$-direction has a channel-shaped form. The superposition of both components yields a bed trough which is symmetric in the $y$-direction (symmetry axis $y = 0$). While the main ice flow is in $x$-direction (from the interior through the bed trough towards the ocean) there is also a flow component in $y$-direction, i.e., from the channel's lateral ridges down into the trough. Resulting convergent flow and associated horizontal shearing enable the emergence of buttressing. Ice is cutoff from the ice shelf and thus calved into the ocean beyond a fixed position $x_{\mathrm{cf}} = 640 \, \mathrm{km}$. During the model spinup no sub-ice-shelf melting is applied. The simulations are carried out using a horizontal resolution of $1 \, \mathrm{km}$.

While the model spinup is closely along the lines of the MISMIP+ experiments, the design of the perturbation experiments is different in this study. Starting from the steady-state ice-sheet-shelf system, basal melting is introduced close to the grounding line in either the central ice-stream region or the lateral shear-margin region(s) of the ice shelf. Ice-stream melting (IS) is confined to the center of the ice shelf, where the ice stream crosses the grounding line (continuous contours in Fig. 3). Shear-margin melting is applied to either one (SM1) or both (SM2) of the two shear margins of the ice-shelf bay, where the ice flows from the ridges into the ice shelf (dotted contours in Fig. 3). That is, the SM1 case applies melting only in region $SM_L$, while in the SM2 case melting takes place in regions $SM_L$ and $SM_R$. In each of the three experiments IS, SM1 and SM2 the melt perturbation is applied over an area of the same length $l = 21$ km along the grounding line. The width $w$, i.e., the extent of the perturbation area into the ice shelf (in $x$-direction for IS and in $y$-direction for SM1/SM2, respectively) is varied between 2 and 16 km in different simulations. This allows us to compare between very confined (small $w$) and more distributed (large $w$) melt patterns while keeping the total sub-shelf mass flux rate $P$ constant. In a further set of experiments $P$ is varied between 0.5 and 2 Gt/yr to investigate the influence of the total melt magnitude.

Throughout all experiments, the resulting local melt rates range between $\approx 1$ m/yr ($P = 0.5$ Gt/yr, $w = 16$ km in experiment SM2) and $\approx 52$ m/yr ($P = 2$ Gt/yr, $w = 2$ km in experiments IS and SM1); see Table 2. To put the magnitudes of the applied perturbations into context we can assume that in the simplest case (1) sub-ice-shelf melt rates are approximately linearly correlated to ocean temperatures, increasing by $10\,\mathrm{m\,yr^{-1}}$ for each Kelvin, as estimated by Rignot and Jacobs (2002), and (2) ocean temperatures increase by about 0.1 to 0.3 K per decade (supported by evidence in Schmidtko et al., 2014). Extrapolating this trend into the near future yields a possible increase of meltrates of several $10\,\mathrm{m\,yr^{-1}}$ within this century, which is consistent with the range of local meltrate perturbations applied here.

The location of the melt area is determined at each model time step and hence adapts to grounding-line movement. It excludes the first floating grid cells directly downstream of the grounding line to assure that the driving stress upstream of the grounding line is not changed by the perturbations. In the IS experiments the location of the perturbation area is symmetric with respect to the setup centerline. In the SM1/SM2 experiments, the $x$-location of the perturbation area also adapts to the length of the confined part of the ice shelf, which we calculate from the $x$-location of the grounding line at the center ($y = 0$ km), $x_{c0}$, and at the margins ($y = \pm 40$ km), $x_{c1}$, of the channel setup. The center of the perturbation area is placed at $x = x_{c0} + 0.4(x_{c1} - x_{c0})$ and thus slightly upstream of the half length of the ice-shelf confinement to exclude melting near "fangs" – grounded features between 480 and 510 km in steady state (Asay-Davis et al., 2016; Cornford et al., 2020). The simulations are run for 100 model years. An unperturbed control run is carried out serving as the reference for the calculation of the time-dependent anomalies.

## 2.3 Cumulative flux response number

Based on buttressing flux response number $\theta_B$ from Reese et al. (2018) we here define the cumulative flux response number (cFRN) as the ratio of the time-integrated change in grounding-line flux and the applied perturbation rate, respectively:

$$\mathrm{cFRN}(t) = \frac{\int_0^t R(t')dt'}{\int_0^t P(t')dt'},$$

(1)

where $R(t)$ is the flux change integrated over the entire grounding line with respect to the reference run in year $t$ and $P(t)$ is the perturbation strength (applied total basal melt rate in year $t$). The cFRN provides a cumulative measure of the sea-level relevant ice-sheet response that is normalized to the applied perturbation magnitude. Thus, it measures the sensitivity of the ice flux across the grounding line to the applied melt rate, i.e., a larger value of the cFRN means that the same perturbation magnitude causes more grounded mass loss. If its value would be one, then the cumulatively perturbed ice mass translates into the same amount of grounding-line flux increase and thus grounded ice loss. A value of zero would occur in an unbuttressed situation, where melting of the ice shelf does not affect the grounding-line flux at all.

## 3 Results

The spun-up ice-sheet-shelf system is characterized by a fast, $\sim 50$ km wide, ice stream that accelerates towards the ice shelf (Fig. 2), being sharply confined by the lateral bed topography (Fig. 1). The strong buttressing force of the ice shelf inside the confinement allows for a stable central grounding-line position on the retrograde slope section (black contour line around $y = 0$ in Fig. 1).

In the perturbation experiments the buttressing is reduced as the applied sub-ice-shelf-melting thins the ice shelf locally. This causes an increase in ice discharge across the grounding line (represented by the cFRN in Fig. 4), accompanied by thinning (Fig. 5) and speed-up (Fig. 6) of the grounded portion of the ice sheet, inducing grounding-line retreat. These effects occur regardless of the location of the perturbations applied in this study. However, the magnitude of the ice-sheet response differs between the three types of experiments, as shown by the cFRN.

Comparing experiments IS and SM2 for the same applied perturbation magnitude (total basal melt rate $P = 2$ Gt/yr) at the end of the perturbation period, the response to the SM2 perturbation is generally stronger regarding the ice-flux increase across the grounding line (Fig. 4; compare dashed to continuous lines), the ice thinning (Fig. 5), ice-flow acceleration (Fig. 6) and grounding-line retreat (gray vs. black contours in Figs. 5 and 6). Though the SM2 perturbation removes the same amount of mass from the ice shelf the induced loss in grounded ice mass, i.e., the sea-level contribution, is about twice as large compared to the IS case. Applying the lateral melt perturbation at only one side of the ice shelf (experiment SM1) with a total melt rate of $P = 1$ Gt/yr can also be thought of as masking out one of the two melt areas in the SM2 experiment. This perturbation leads to an *absolute* response in terms of grounded-ice acceleration and grounding-line retreat that is similar to the IS case, thus being weaker than in the SM2 case. However, *relative* to the applied perturbation strength the response magnitude is twice as large compared to the IS experiments and on the same order of the SM2 experiments, as can be seen from the cFRN (Fig. 4). In other words, melting at (one of) the ice-shelf shear margins (SM1/SM2) is twice as effective compared to ice-stream melting (IS) as it requires only half of the perturbation strength to induce the same response magnitude.

### 3.1 Physics underlying the enhanced ice-flux sensitivity to shear-margin melting

The primary reason for the different ice-sheet response magnitudes lies in the finding that basal melting in the shear-margin regions has a more sustained effect on local ice-shelf thinning than ice-stream melting, implying a stronger reduction in ice-

shelf buttressing (Fig. S2). The thinning in the SM1/SM2 case has a very localized pattern, being almost entirely confined to the perturbation region(s) (cyan contours in Fig. 5). In fact, at the end of the 100-yr perturbation, the local reduction of the ice-shelf thickness in the shear-margin case can be twice as large as in the ice-stream case (minimal fraction of original ice thickness, $f_{\min}$, stated in lower left corner of each panel in Fig. 5). This is due to the fact that the steady-state ice flow from 1) the lateral ridges (where the ice is very stagnant) and from 2) the direction of the ice divide into the lateral perturbation areas is comparatively low (Figs. 2 and S1a). In contrast, inside the bed trough there is strong ice advection from the ice-sheet interior towards the grounding line.

The effect of the melt location on the thinning pattern can be illustrated by the following example: Assume a melt rate of $1$ m/yr applied to an area of $1$ km$^2$ in the IS region and in the SM regions, respectively. The steady-state ice flow speed is $\sim 500$ m/yr in the IS region and $\sim 100$ m/yr in the SM regions. Now let's take the Lagrangian perspective of an ice column traveling through the melt regions. In the IS case the column spends two years in the melt region and is thus melted by $2$ m in total after it leaves the melt region. In contrast, in the SM case the ice column spends five years in the melt region and is thus melted by $5$ m in total after leaving the melt region. Going back to the Eulerian perspective, ice in the IS region is thus losing maximally $2$ m of thickness, while ice in the SM region is thinning by up to $5$ m. This means that ice-shelf thinning is less concentrated in the center of the trough, where the ice stream is fastest (red central longitudinal stripe in Fig. 5a). Here the thinning is transported downstream, where it has less effect on buttressing (Reese et al., 2018; Zhang et al., 2020). The strong decline of the ice stream's speed/flux towards the trough's lateral margins results in a more concentrated ice-shelf thinning pattern at the shear margins.

The simulated enhanced thinning in the shear-margin regions is further promoted by the orientation of the applied melt perturbation stripes that have a length-to-width ratio of $l/w > 1$. In the SM case the ice flows along the long edge of the melt area due to its orientation in along-flow direction. This increases the exposition time to melting of an ice column passing through the perturbation region (passage distance $l$) and thus increases the thinning magnitude in the melt area compared to the IS case in which the ice crosses the short edge of the perturbation area (passage distance $w$). In fact, the gap in the response sensitivity between the IS and SM cases reduces if the melt-strip length $l$ is reduced, as shown by additional experiments that apply a halved melt strip length of $l = 11$ km (Fig. S9). Nevertheless, the qualitative result of an enhanced long-term response to shear-margin melting compared to ice-stream melting remains the same, also for length-to-width ratios of $l/w \approx 1$ or $l/w < 1$ (green and orange curves in Fig. S9b). This suggests that the strong difference of the ice-flow speed between the central ice stream and the lateral shear margins (as described in the previous paragraph) is the dominant cause for the response difference.

The perturbations effect that the fast streaming of the grounded ice inside the bed trough intensifies near the grounding line (Fig. 6). Despite of deviations in the ice-flow response between the different perturbation experiments, the general flow pattern of the grounded ice stream is not altered (compare Figs. S1b-e). Induced ice-flow changes in the regions of the lateral ridges outside the ice stream are negligible. As a result, the ice flow remains strongest (weakest) in the central part (the lateral margins) of the ice stream, regardless of where the perturbation is applied. Hence, the enhanced susceptibility of the ice-shelf shear zones to basal melting, as described in the previous paragraph, is maintained throughout the perturbation period.

While almost the entire ice shelf accelerates in the SM1/SM2 experiments (local speed-up of over $100\mathrm{m/yr}$; reddish colors in Fig. 6b,c), in the IS experiment the major ice-shelf part downstream of the perturbation area slightly decelerates (Fig. 6a) due to the thinning-induced weakening of the driving stress in main flow ($x$-) direction. This leads to reduced advection out of the central perturbation area and thus less thinning there. The time evolution of the melt-induced thinning and speed-up patterns is visualized in Figs. S6 and S7.

## 3.2 Role of duration, area and magnitude of the perturbation

Above we saw for an exemplary perturbation strength of $P = 2\,\mathrm{Gt/yr}$ (experiments IS and SM2) and $P = 1\,\mathrm{Gt/yr}$ (experiment SM1) that shear-margin melting can be twice as efficient as ice-stream melting (Fig. 4). We here conduct all three perturbation experiments for different widths $w$ of the basal-melt strip(s) and different perturbation strengths (total melt rates $P$), which are given in Table 1. The associated ranges of the size of the perturbation areas and of the resulting local melt rates are listed in Table 2. To quantify the difference in the response magnitude for all ice-stream and shear-margin melt perturbations, respectively, we calculate the ratio of the cFRN values, i.e., $r_{\mathrm{SM1}} = \frac{\mathrm{cFRN_{SM1}}}{\mathrm{cFRN_{IS}}}$ and $r_{\mathrm{SM2}} = \frac{\mathrm{cFRN_{SM2}}}{\mathrm{cFRN_{IS}}}$. For the major part of the perturbation period, shear-margin melt induces a stronger response than ice-stream melt (exception for the lowest total melt rate $P = 0.5\,\mathrm{Gt/yr}$ and the widest melt-strip width $w = 16\,\mathrm{km}$) and it thus holds $r_{\mathrm{SM1}}, r_{\mathrm{SM2}} > 1$, with a peak occurring during the first decades (Fig. 7). We also find that in general $r_{\mathrm{SM1}} > r_{\mathrm{SM2}}$ (Fig. S8). The results show that higher local melt rates (small $w$ and/or large $P$) favor higher cFRN ratios. They stabilize at values of up to 2.5 towards the end of the perturbation period. However, within the first few model years the majority of the experiments shows larger cFRN values in the ice-stream case than in the shear-margin case, i.e., $r_{\mathrm{SM1}}, r_{\mathrm{SM2}} < 1$ (insets in Figs. 7 and S8 show the first 20 model years). This ratio reverts in most cases after about five to ten model years, with high local melt rates favoring a faster transition. Within this short initial period, the response is close to the diagnostic response which is higher for a perturbation in the central parts (Reese et al., 2018; Zhang et al., 2020). But the system evolves and, as discussed in the Sec. 3.1, the IS thinning is smeared out over time across the ice shelf due to the fast advection. Lower advection rates in the SM regions in contrast induce stronger local thinning in more buttressing relevant regions.

Overall, the magnitude of the simulated ice-sheet response increases with decreasing $w$. This can be seen from the deviations in the grounding-line flux increase (Fig. S4), the buttressing reduction (Fig. S3), and the magnitude of grounding-line retreat (Fig. S5). Fig. 8 shows the maximum cFRNs which support this. A reduction of $w$ under a fixed $P$ value increases the local melt rates (each halving of $w$ doubles the local melt rate) and thus causes higher ice-shelf thinning rates close to the grounding line. This leads to a larger buttressing reduction, explaining the larger increase in grounded ice loss and grounding-line retreat. However, in case of a low perturbation strength and a small melt-strip width the above relation does not apply for the IS experiments where differences in the cFRN between $w = 2$ and $4\,\mathrm{km}$ are small or even reversed (light and dark blue squares in Fig. S3-S5). The general increase in the cFRN with declining $w$ is much stronger in the SM1/SM2 experiments than in the IS experiments. This is due to the fact that a reduction in $w$ concentrates the basal melting closer to the grounding line, i.e., in the SM1/SM2 experiments the melting is shifted towards the stagnant lateral ice-shelf margins. There the upstream ice supply is sparse, leading to enhanced local thinning rates and, in turn, a stronger ice-sheet response. For the three largest applied

shear-margin melt rates (simulations SM1 with $w = 2$ km for $P = 1.5$ Gt/yr and $w = 2$ km / $w = 4$ km for $P = 2$ Gt/yr) the thinning is intense enough to locally reduce the ice-shelf thickness to zero (cut-off curves in Fig. S8). Due to the lack of comparability these experiments are excluded from the analysis (no data points in Figs. 8 and S3-S5).

The spread in the cFRN under a variation in $P$ for a given $w$ is much larger in the IS experiments (standard deviations in the cFRN, $\sigma_{\text{cFRN}}$, range from around 0.4 and 0.6) than in the SM1/SM2 experiments ($\sigma_{\text{cFRN}}$ between 0.07 to 0.12). A large spread indicates a non-linear response of the grounded ice to different perturbation strengths. In the ice-stream case the lowest perturbation strength ($P = 0.5$ Gt/yr) is by far the most efficient one, yielding the largest cFRN value regardless of the melt-strip width. The same applies to the shear-margin melt patterns but only for the medium/wide melt-strip widths $w = 8$ and $16$ km. For smaller $w$ there is no optimal $P$ and the cFRN values lie close to one another.

## 4  Discussion and conclusions

Carrying out idealized numerical simulations we investigate the transient response of an inherently buttressed marine ice-sheet-shelf system (Figs. 1, 2 and S1a) to basal melt perturbations that are applied close to the grounding line in the central ice-stream (IS) region and the lateral shear-margin (SM1/SM2) regions of the ice shelf (Fig. 3). The applied perturbations thin the ice shelf (Fig. 5) and thus reduce its buttressing strength (Figs. S2 and S3), inducing an increase in ice discharge across the grounding line (Figs. 4). Our analysis reveals that the flux response strongly depends on the duration, the location, the extent and the strength of the perturbation:

1) The initial change in grounding-line flux (within a few years) is slightly larger for the case of ice-stream melting compared to shear-margin melting (insets of Fig. 7). This is in line with results from Reese et al. (2018) who find strongest instantaneous responses in the grounding-line flux for thinning directly downstream of the grounded Antarctic outlets (comparable to the central ice-stream melt region here). Diagnostic experiments based on the same topographic setup as used in our study show a similar instantaneous response to melting in the ice stream and along the shear margins, respectively (Fig. 2 of Zhang et al. (2020)).

2) For continued melting (more than the initial five to ten years) the flux response becomes significantly stronger with respect to shear-margin melt, being up to 2.5 times as large in the SM1/SM2 case compared to the IS case after 100 model years (Fig. 7). Accompanying induced changes in the upstream grounded ice, i.e., flow acceleration (Fig. 6 and S7) and thinning (Fig. 5 and S6), as well as grounding-line retreat (Fig. S5), are much more pronounced under shear-margin melting. We attribute this to the flow characteristics of the simulated ice-sheet-shelf system. Due to the nature of such a channelized ice stream, the major portion of the ice discharge across the grounding line occurs in the central part of the bed trough, i.e., the fast-flowing ice stream (Figs. 2 and S1a). As a result, the faster advection of ice through the ice stream center spreads the thinning signal across the shelf (Fig. 5a) into less buttressing relevant regions (Reese et al., 2018; Zhang et al., 2020). Contrary, in the SM1/SM2 experiments the bulk of the thinning remains confined to the shear-margin perturbation regions (Fig. 5b,c). This way a comparatively strong and localized shear-margin thinning pattern emerges. The thinning is further facilitated by the alignment of the applied melt perturbations along the grounding line in a stripe-type shape (length-to-width ratios $l/w > 1$),

which is a common feature of observed ice-shelf melt patterns (Dutrieux et al., 2013; Marsh et al., 2016; Alley et al., 2016; Shean et al., 2019). Diagnostic studies find that the grounding-line flux response is an increasing function of ice-shelf thinning (Reese et al., 2018; Gudmundsson et al., 2019), consistent with the above-described increase of the signal from shear-margin melting over the one from ice-stream melting with time.

3) A stronger response is generally favored by a strong perturbation (large $P$) and a high concentration of melting close to the grounding line (small $w$; Fig. 8). Comparing confined to distributed melting in our simulations for a given perturbation strength reveals a smaller increase in ice discharge across the grounding line (Fig. S4) and less grounding-line retreat (Fig. S5) in the case of more distributed melting. This is in agreement with results from idealized flowline simulations of a buttressed ice-sheet-shelf system by Gagliardini et al. (2010). They find grounding-line advance accompanied by volume gain when reducing

the concentration of sub-ice-shelf melting to the grounding line, while leaving constant the total amount of melted ice.

## 4.1   Further possible shear-margin effects and model limitations

The isothermal approach taken here neglects the effect of shearing-generated heating and thus ice softening within the ice-stream shear margins, which is a common feature of Antarctica's fast flowing ice streams (Suckale et al., 2014; Perol and Rice, 2015; Meyer and Minchew, 2018). Accounting for such softening in our simulations would likely result in a faster steady-state

ice stream due to the reduced amount of buttressing provided by the weaker shear margins, also affecting the steady-state grounding-line position and ice geometry. An associated larger ice advection into the ice shelf would reduce the magnitude of melt-induced ice-shelf thinning. At the same time, diminished ice-shelf buttressing in steady state could increase the sensitivity of the response to a further buttressing reduction in the course of the basal-melt perturbations. The acceleration of the ice stream in response to the perturbations enhances horizontal shearing in the lateral ice-shelf margins in the SM melt case (Figs. 6 and

S7) and thus would induce further shear-margin softening. In simulations of West Antarctica's Thwaites Glacier, which were forced by ice-stream-type melting, a softening of the shear margins promoted higher grounding-line retreat rates (Joughin et al., 2014). Since viscous heating can produce water-saturated, temperate ice it plays an important role regarding the localization of the ice-stream shear margins if topographical constraints of the underlying bed are weak (Jacobson and Raymond, 1998; Haseloff et al., 2019; Hunter et al., 2021). Moreover, temperate ice in the grounded ice-stream shear zones may provide input of

subglacial freshwater into the ice-shelf cavity. There it can promote buoyancy-driven plumes that cause enhanced, channelized melting beneath the ice-shelf shear margins (Jenkins, 2011; Marsh et al., 2016). In our simulations, the location of the shear margins of the ice-sheet-shelf system is determined by the pronounced channel-type bed topography. Due to the fixed width of the prescribed bed trough, also the width of the steady-state ice-stream shear margins is constant throughout the experiments. We illustrate the potential influence that a variation of the shear-margin width would have in our experiments by the following

example: a wider (narrower) shear margin would imply a flatter (steeper) gradient of the ice-speed profile (in $y$-direction) within the bed through, as long as the centerline ice speed remains unchanged. This would come along with weaker (stronger) advection in the regions close to the lateral confinement, i.e., the SM melt regions, increasing (decreasing) local melt-induced ice-shelf thinning and thus leading to higher (lower) grounding-line flux sensitivities to the lateral melt perturbations.

Our simulations do not account for the process of ice-shelf fracturing (Schulson and Duval, 2009). Basal melting in ice-shelf shear margins that are prone to fracture-induced mechanical weakening can amplify the fracturing and thus diminish the ice shelf's backforce in addition to the purely thinning-induced buttressing reduction (Shepherd, 2003; Borstad et al., 2016; Goldberg et al., 2019). In fact, shear-margin weakening is suggested to have the potential to initiate a positive feedback loop between damage in ice-shelf shear margins, speedup, further shearing and thus enhanced weakening of the margins (Alley et al., 2016, 2019; Lhermitte et al., 2020). Satellite observations indicate a rapid development of damage areas in the shear zones of West Antarctica's Pine Island and Thwaites glaciers (MacGregor et al., 2012; Lhermitte et al., 2020; Wild et al., 2022). Using the MISMIP+ setup in simulations that involved a continuum damage model, Lhermitte et al. (2020) highlighted the importance of the damage feedback, leading to, e.g., enhanced grounding-line retreat. Their study included simulations that introduced basal shear-margin melting on top of melting in the central ice-stream region, finding that, in terms of grounding-line retreat, the effect of shear-margin weakening due to the additional lateral melting is comparable to quadrupling the central melting. Our approach thus underestimates the response to basal shear-margin melt compared to simulations that would include damage and/or viscous heating. Further variables whose analysis is beyond the scope of our study include the length and the width of the ice-shelf embayment as well as the ice rheology. All of these parameters substantially influence the buttressing strength of the ice shelf (Dupont and Alley, 2005; Goldberg et al., 2009; Gudmundsson et al., 2012) and thus their variation might alter our results.

There are several simplifications in the design of the model setup and the experiments (shallow stress balance, idealized bed topography and perturbation, fixed calving front), thus reducing complexity of modeled ice flow. The synthetic bed topography and the idealized forcing used here aim at a conceptual understanding of the ice-sheet response to the applied perturbation in contrast to the attempt of investigating a real-world system that would include a much wider range of physical effects. For instance, the smooth bed geometry used here does not account for bumps usually found in observations of the sub-glacial topography and which would interfere with grounding-line dynamics (e.g., Alley et al., 2007; Favier et al., 2012). Also, the distribution of the sub-shelf melting in space and their evolution in time would be much more complex in a real-world system (e.g., Dutrieux et al., 2013) in contrast to the spatially very confined, step-like perturbations applied in our simulations. However, the approach taken here allows for an analysis of the first-order effects on ice-shelf buttressing, ice discharge and grounding-line migration. The simplicity of the applied perturbations facilitates the differentiation of the mechanisms underlying the ice-sheet response to ice-shelf thinning in the ice-stream and shear-margin grounding-line regions, respectively. A spatially refined application of such perturbations to Antarctica's buttressed outlet glaciers in transient simulations would allow to systematically investigate their time-dependent sea-level response, providing valuable insight beyond the findings of the present study.

## 4.2 Importance of results for real-world systems

Our simulations are based on a synthetic setup that is designed to represent an Antarctic-type outlet glacier, such as Pine Island Glacier (Asay-Davis et al., 2016; Lhermitte et al., 2020). Within the limits of such a simplified representation, our findings underline the important role of ice dynamics in the regions adjacent to the grounding line (grounded and floating regimes)

interacting with enhanced sub-ice-shelf melting to regulate grounded mass loss. In particular, our results suggest that the dynamics of fast, marine, outlet glaciers that are buttressed by a laterally confined ice shelf - a configuration that is often found in Antarctica - are particularly susceptible to melting in the stagnant, but buttressing-relevant parts of their ice shelves. These regions could include lateral ice-stream margins (as in our simulations) or the vicinity of ice rises. Observational evidence

for the occurrence of elevated sub-ice-shelf melting in such regions exists at least occasionally around the Antarctic Ice Sheet (Alley et al., 2019), e.g., for Pine Island Ice Shelf (Shean et al., 2019) and Crosson/Dotson ice shelves (Goldberg et al., 2019) in the Amundsen Sea in West Antarctica, Roi Baudouin Ice Shelf, East Antarctica (Berger et al., 2017) or Filchner-Ronne Ice Shelf (Adusumilli et al., 2020). According to our results, melting in such regions does not have to be widespread and intense but can be relatively localized and moderate to induce a stronger ice-shelf buttressing reduction (and thus sea-level response),

compared to melting in the faster moving, central streaming parts of an ice shelf, where the strongest present-day melt rates are usually observed. Consequently, occurrences of continued melting/thinning beneath shear zones of Antarctic ice shelves such as the ones mentioned above should receive special attention. Observed heat-induced and damage-induced shear-margin weakening involves further processes that increase the ice-shelf/ice-sheet sensitivity to shear-margin melting. The physical mechanism studied here and its implications for global sea-level rise might gain importance in the future as sub-ice-shelf melt

rates are expected to increase under continuing global warming.

**Table 1.** Parameters and their values varied throughout the experiments.

| Parameter | Value | Unit | Physical meaning |
|-----------|-------|------|------------------|
| $a$ | 0.3 | $\mathrm{m\,yr^{-1}}$ | Surface accumulation rate |
| $A$ | $8 \cdot 10^{-25}$ | $\mathrm{Pa^{-3}\,s^{-1}}$ | Ice softness (Glen's flow law coefficient) |
| $\beta^2$ | $3.16 \cdot 10^6$ | $\mathrm{Pa\,m^{-1/3}\,s^{1/3}}$ | Basal friction coefficient in Weertman law (Asay-Davis et al., 2016, Eq. 6) |
| $m$ | 3 | | Basal friction exponent in Weertman law (Asay-Davis et al., 2016, Eq. 6) |
| $l$ | 21 | km | Length of perturbation area |
| $P$ | $\{0.5, 1.0, 1.5, 2.0\}$ | $\mathrm{Gt\,yr^{-1}}$ | Total melt rate in perturbation area |
| $w$ | $\{2, 4, 8, 16\}$ | km | Width of perturbation area |
| $x_{\mathrm{cf}}$ | 640 | km | Position of fixed calving front in right-hand half of domain |

**Table 2.** Ranges of perturbation areas and melt rates for each of the three perturbation experiments.

| | **IS** | **SM1** | **SM2** |
| | (Ice-stream melting) | (One-sided shear-margin melting) | (Two-sided shear-margin melting) |
|---|---|---|---|
| Area of perturbation ($\mathrm{km^2}$) | $42 - 336$ | $42 - 336$ | $84 - 672$ |
| Applied total melt rates ($\mathrm{Gt\,yr^{-1}}$) | $0.5 - 2$ | $0.5 - 2$ | $0.5 - 2$ |
| Local melt rates ($\mathrm{m\,yr^{-1}}$) | $\approx 2 - 52$ | $\approx 2 - 52$ | $\approx 1 - 26$ |

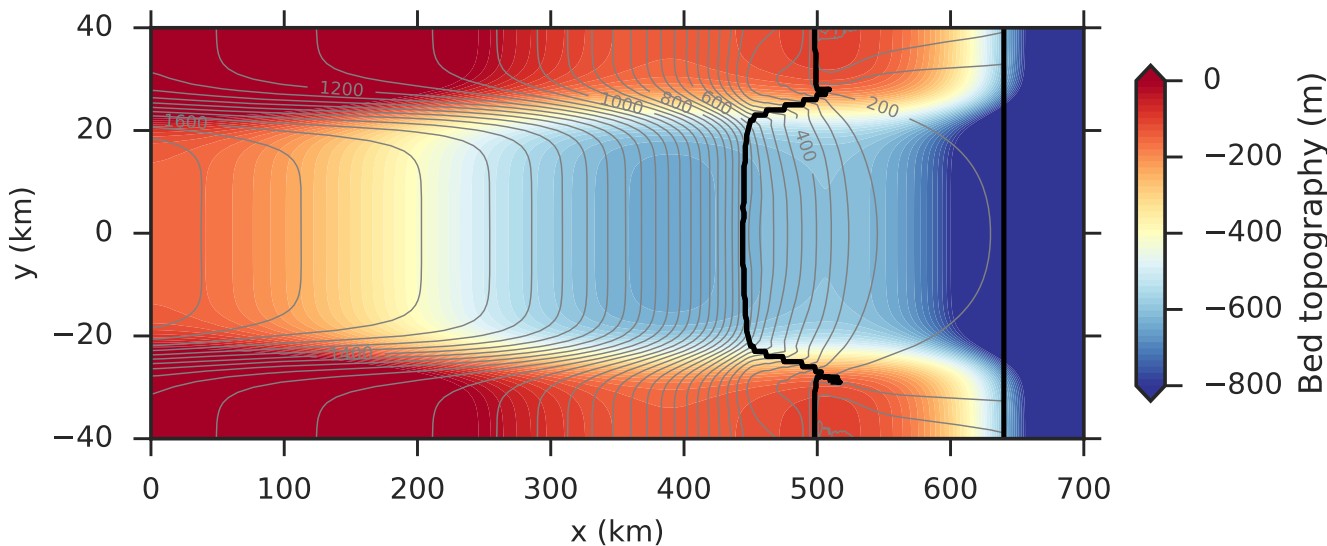

**Figure 1.** Top view of channel-type bed topography (Asay-Davis et al., 2016; Cornford et al., 2020) used in this study. It is characterized by an overdeepening (retrograde bed section) in $x$-direction on which the grounding line (black contour) of the spun-up ice-sheet-shelf system has a stable steady-state position. Gray contours represent ice thickness in 50-m steps. The fixed calving front is shown by the straight black contour at $x = 640$ km.

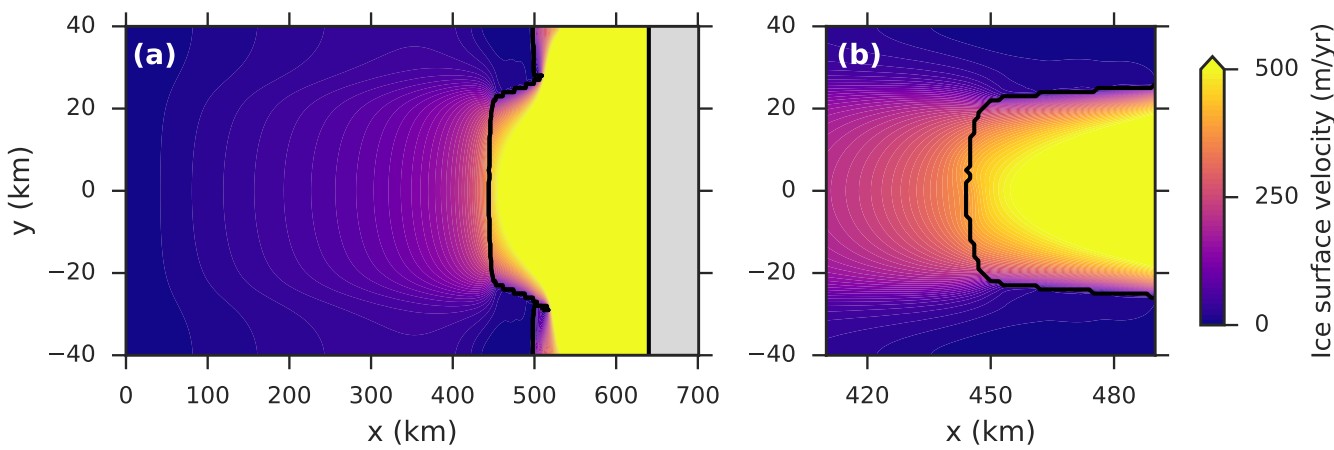

**Figure 2.** Steady-state ice surface speed for **(a)** the entire model domain and **(b)** the grounding-line region. Grounding line and calving front represented by black contours.

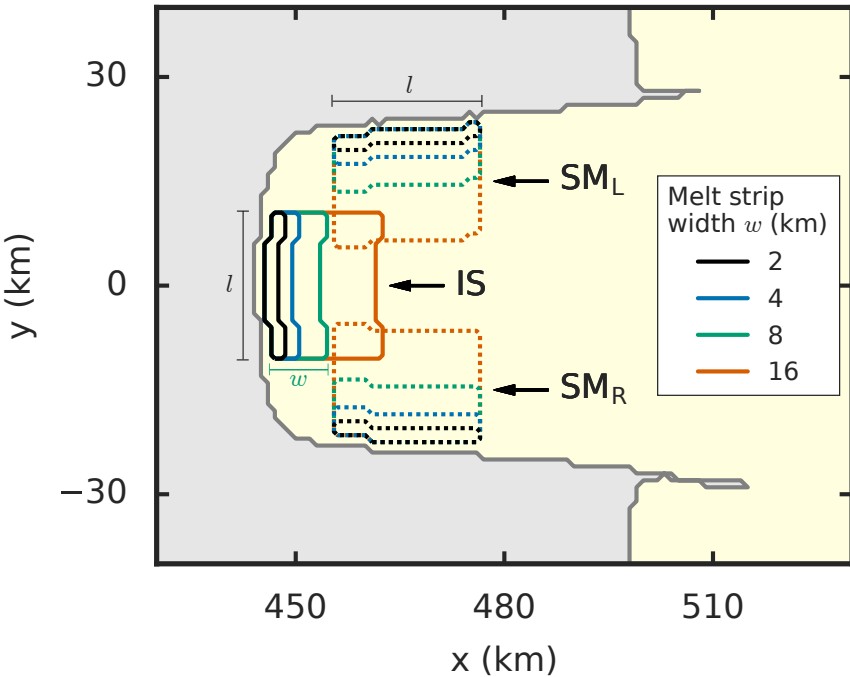

**Figure 3.** Ice-shelf regions IS, $SM_L$ and $SM_R$ in which the basal melt perturbations are applied (zoom into grounding-line region). While all perturbation areas have the same length $l$ along the grounding line, their width $w$ is varied in the simulations (colored contours). The steady-state grounding line is represented by the dark grey contour, the ice sheet is colored light gray, the floating ice shelf is colored yellow. Continuous and dotted contours are used for better distinguishability between the central and lateral melt regions.

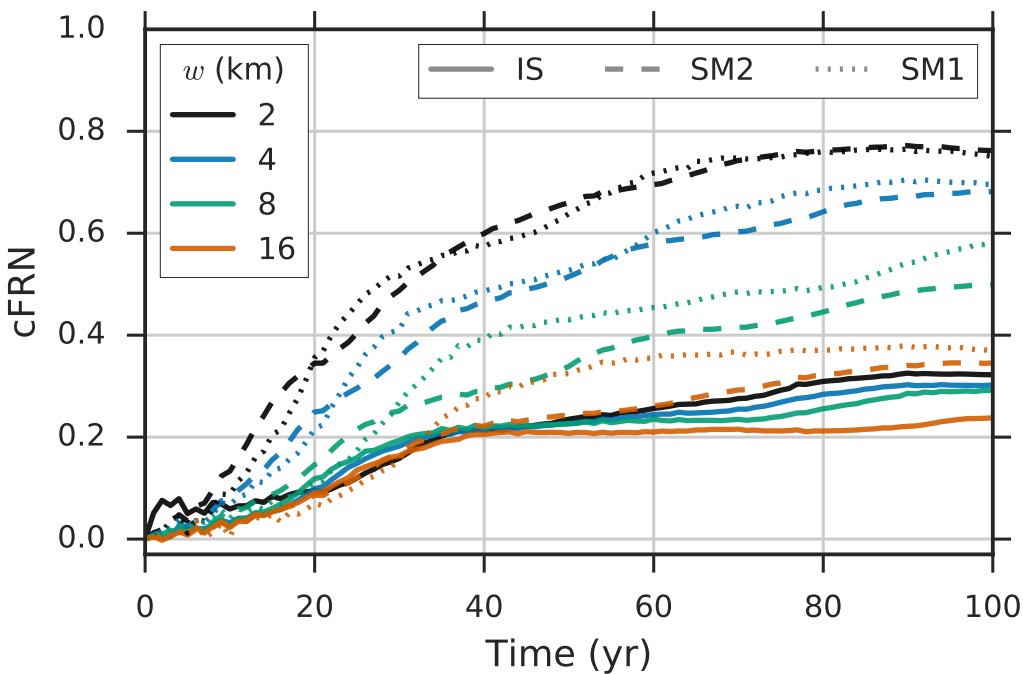

**Figure 4.** Time evolution of the cumulative flux response number, cFRN (Eq. 1), for the three different perturbation experiments and the four applied melt-strip widths $w$. Note that the total melt rate $P$ is $2\ \mathrm{Gt/yr}$ in the IS and SM2 cases and $1\ \mathrm{Gt/yr}$ in the SM1 case for a better comparability between the SM1 and SM2 cases.

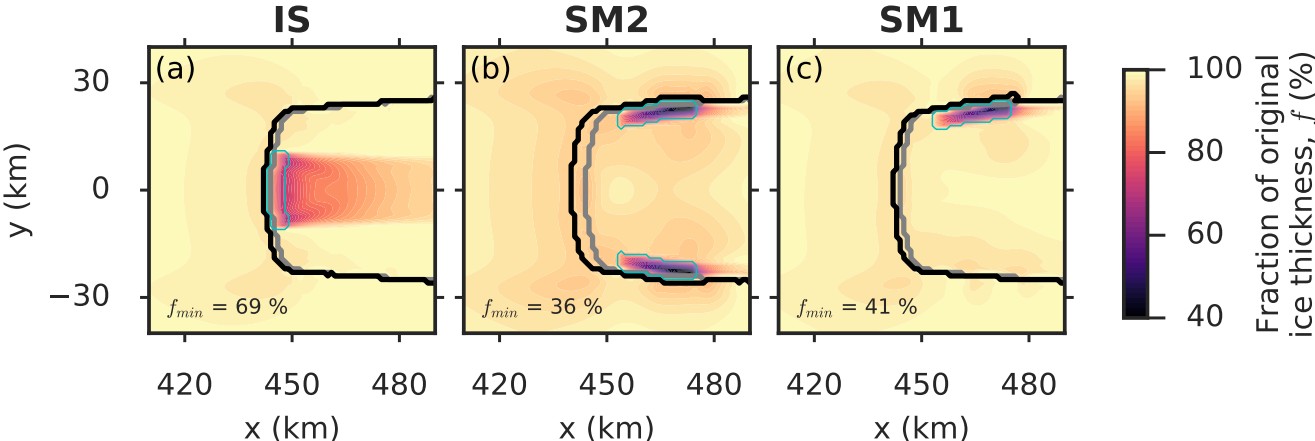

**Figure 5.** Fraction of initial ice thickness $f$ (colorbar) in the vicinity of the grounding line at the end of the 100-yr perturbation for the three different perturbation types and an applied melt-strip width of $w = 4$ km. In each panel the minimum value of $f$ is given in the lower left corner. Thick contours represent the grounding-line position in the initial state (grey) and in the perturbed states (black). The thin cyan contour denotes the perturbation area. Note that the total melt rate $P$ is 2 Gt/yr in the IS and SM2 cases and 1 Gt/yr in the SM1 case for a better comparability between the SM1 and SM2 cases. The thinning pattern is shown for further time slices in Fig. S6.

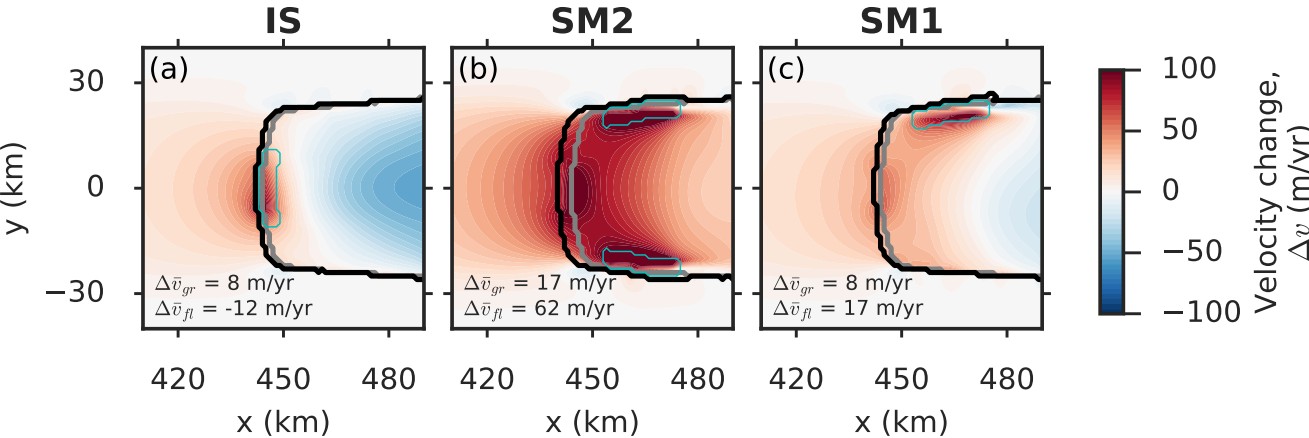

**Figure 6.** Change in ice speed $\Delta v$ (colorbar) in the vicinity of the grounding line at the end of the 100-yr perturbation for the three different perturbation types and an applied melt-strip width of $w = 4$ km. In each panel the spatial mean of the grounded and floating speed changes (average over the displayed area), $\Delta \bar{v}_{gr}$ and $\Delta \bar{v}_{fl}$, respectively, are given in the lower left corner. Thick contours represent the grounding-line position in the initial state (grey) and in the perturbed states (black). The thin cyan contour denotes the perturbation area. Note that the total melt rate $P$ is 2 Gt/yr in the IS and SM2 cases and 1 Gt/yr in the SM1 case for a better comparability between the SM1 and SM2 cases. The pattern of changing ice-speed is shown for further time slices in Fig. S7.

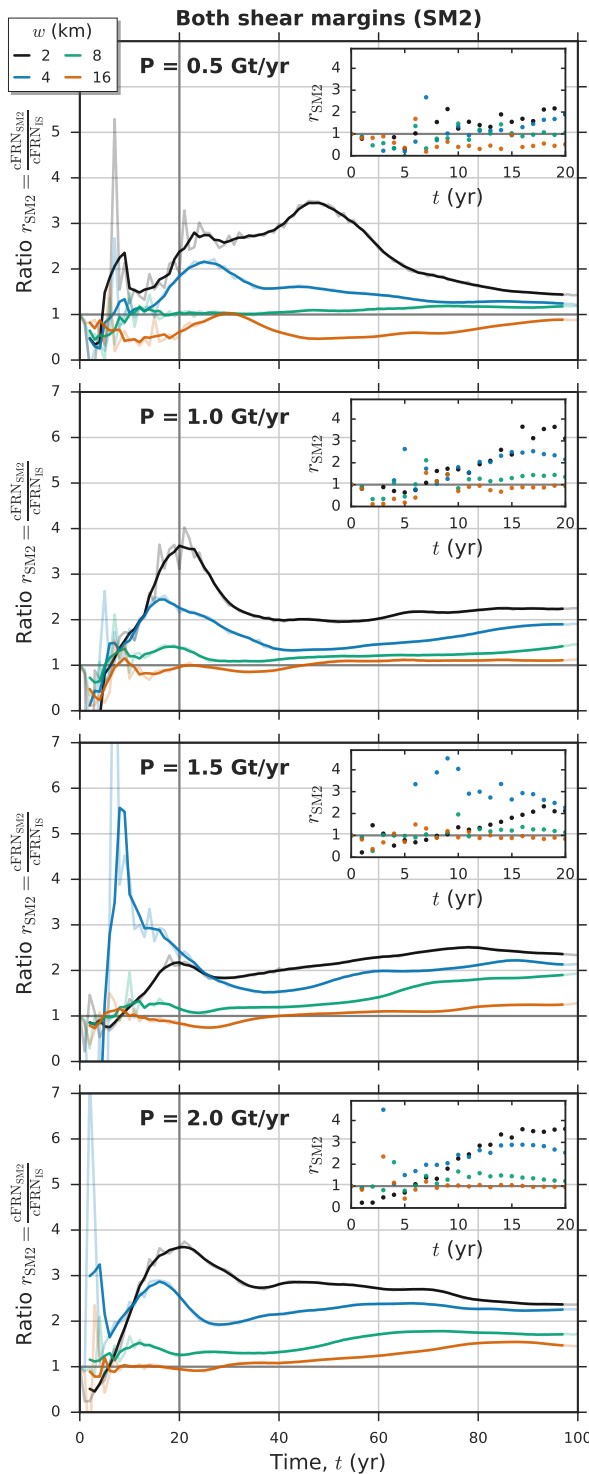

**Figure 7.** Time evolution of the cFRN ratio $r_{SM2}$ for the four perturbation strengths $P$ and the four melt strip widths $w$ (colors given in the legend). The curves show the 5-year running mean of the yearly data (light colors). For each panel the yearly data points for the first 20 model years are shown in the corresponding inset. A comparison to the qualitatively very similar curves of $r_{SM1}$ is given in Fig. S8.

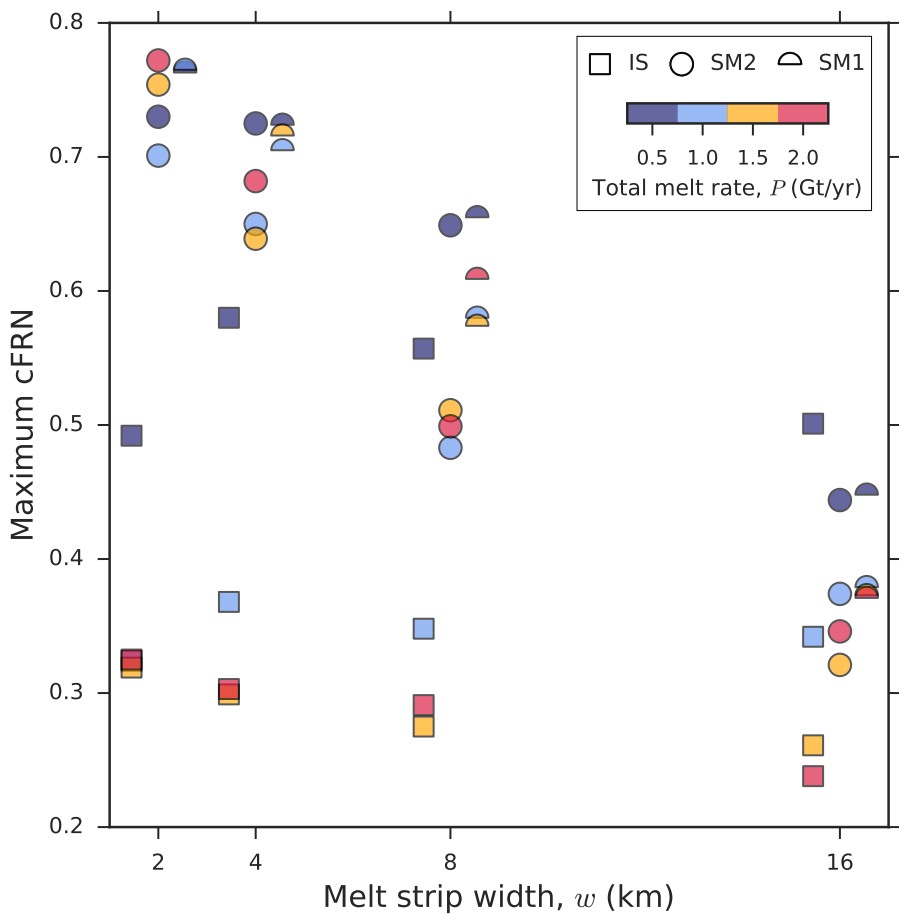

**Figure 8.** Maximum of the cFRN dependent on the melt-strip width $w$ ($x$-axis) and perturbation strength $P$ (colorbar). The perturbation types are represented by individual symbols (legend). For better visibility the data points of the three perturbation types are slightly shifted against each other on the $x$-axis.

*Code and data availability.* The model code used in this study is based on PISM stable version 1.0 and can be obtained from https://doi.org/10.5281/zenodo. Simulation data and plotting scripts are available from the PANGAEA repository (https://www.pangaea.de).

*Author contributions.* The design of this study involved all authors. RR developed the model code for the basal-melt perturbations. JF performed the numerical simulations. JF prepared the figures and the manuscript with contributions from RR. All authors commented on the manuscript.

*Competing interests.* The authors declare no conflict of interest.

*Acknowledgements.* This work was supported by the Deutsche Forschungsgemeinschaft (DFG) in the framework of the priority programme "Antarctic Research with comparative investigations in Arctic ice areas" SPP 1158 through grant WI 4556/6-1. RR and RW are grateful for support by the European Union's Horizon 2020 research and innovation programme under Grant Agreement No. 820575 (TiPACCs). RW further acknowledges support by the European Union's Horizon 2020 research and innovation programme under Grant Agreement No. 869304 (PROTECT). Development of PISM is supported by NASA grant NNX17AG65G and NSF grants PLR-1603799 and PLR-1644277. The authors gratefully acknowledge the European Regional Development Fund (ERDF), the German Federal Ministry of Education and Research and the Land Brandenburg for supporting this project by providing resources on the high performance computer system at the Potsdam Institute for Climate Impact Research. We thank three anonymous referees for their valuable comments and suggestions which helped to improve the manuscript.

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
