# Peer review of "Shear-margin melting causes stronger transient ice discharge than ice-stream melting in idealized simulations"

_The Cryosphere, 2021_

## Author Comment (AC1)

*The focus of this paper is on the sea-level rise response from localized melting on regions of a buttressing ice shelf. The melting is applied either at the grounding line or along the lateral edges where the topography increases and the downstream flow is slower (Figures 2 & 3), i.e. the shear margins. The difference between the effects of additional melting at the grounding line versus melting below the ice shelf shear margins is notable. And it make sense from a force balance perspective, thinning the shear margin lowers the buttressing balance and the ice stream will accelerate. Similarly, if we considered a unbuttressed ice shelf with a single pinning point, it would be clear that melting at the pinning point would affect the flow more than melting at the grounding line. Although it is an intuitive result with few actionable consequences, I would tepidly support publication in The Cryosphere.*

We would like to thank Referee#1 for their willingness to review our manuscript, the helpful comments and the constructive criticism. We are glad for the referee's positive assessment of our study and are happy to hear that they would support the publication in TC. We are gladly willing to implement all the suggestions and points raised by the referee (see below).

*Additional thoughts:*

- *the force balance argument described above doesn't appear in the text and the description of the difference between the grounding line and shear margin melting is too thin.*

  We thank the referee for this helpful comment. We will add the mentioned force balance argument and will expand the description of the difference of the two melt patterns as suggested by the referee.

- *I find the 'three dimension' description of the simulations as misleading, since SIA/SSA hybrid can have three-components but is still depth integrated.*

  We are willing to change the wording according to the referee's suggestion.

- *the second sentence in the abstract is missing a comma before `the melting'.*

  Will be corrected.

- *what is solid-ice? I would replace this with 'grounded' both in the abstract, introduction, and anywhere. Right? Solid, as opposed to what?*

  We are glad to change the wording as suggested by the referee.

- *it seems like the SM1 is nearly as effective at instigating ice flux as SM2, yet the text in the second paragraph on page 5 is confusing as compared to Figure 4.*

  We think the confusion here is based on the difference between the absolute ice-flux response (mentioned in lines 11-12 on page 5) and the relative ice-flux response (mentioned in lines 13-15 on page 5 and shown in Fig. 4). We will revise this paragraph for clarification.

- *lastly, it seems like the authors have discovered for themselves why shear margins are important. Yet I know that others have worked on shear margins, such as Lhermitte et al (2020). I suggest a clearer*

*connection to the existing literature.*

We are thankful for this hint and will revise our manuscript in terms of a deeper connection to the existing literature, as the referee suggests

*S. Lhermitte, S. Sun, C. Shuman, B. Wouters, F. Pattyn, J. Wuite, E. Berthier, and T. Nagler. Damage accelerates ice shelf instability and mass loss in Amundsen Sea Embayment. PNAS, 117(40):24735–24741, 2020*

---

## Author Comment (AC2)

*This study evaluates the sensitivity of ice flux from ice streams to the location of sub-ice shelf meltwater. In particular, the authors compare localized sub-ice shelf melting that occurs in the trunk of the ice shelf to melting that occurs in the shear margin, where ice velocity decreases rapidly. In model runs of PISM, they find that localized melting in the shear margins affects ice flux more than melting in the trunk of the ice stream and they suggest that this is due to the slower velocities in the shear margin. The study seems comprehensive and is laid out in an intuitive manner. The paper itself is well-written. I believe there is much to think about when it comes to the effects of shear margin dynamics on ice shelf buttressing, and I am heartened to see studies tackling this question. There are some comments below that may improve the readability and clarity of the paper.*

First of all, we would like to thank Referee#2 for their willingness to review our manuscript. We are grateful for the referee's positive assessment of our study and the helpful comments and suggestions that we are happy to address to improve our manuscript.

*Dynamics: In general, while I follow the logic of the underlying dynamics that cause shear margin melting to affect ice flux more than melting in the trunk, I felt that this argument could have been presented more clearly in the paper. While the discussion section does introduce a number of interesting points, I found it to be missing a clear explanation for the reasons behind the disparity in flux response. There is some explanation in the results section in lines 17-25 of page 5, but I found this explanation to be a bit buried in the results section and quite short given that this appears to be the primary physical explanation for the results of the paper.*

We are grateful for this hint. As suggested by the referee, we will revise the manuscript in order to give the mechanism underlying the flux differences more visibility.

*I also wondered if the study needed more of a formal connection to other shear margin studies that consider the effect of shear margin dynamics on ice shelf/ice stream stability. For example, Alley and others 2019 proposes a physical mechanism for the localization of melt underneath ice shelf shear margins, and invoking these studies would strengthen the motivations of this work quite a bit. Further, there's been quite a bit of work done on heating in shear margins which suggest that shear margins are likely to be quite warm (and even temperate), and I would be interested to know whether this may further increase basal melting in these regions given that the ice is already quite warm (see: Suckale and others 2014, Perol and Rice 2015, Haseloff and others 2019).*

We thank the referee for mentioning these important studies related to the topic of our study that we were missing. We will be glad to reference and discuss these papers in our manuscript.

*Connection with observations and modeling: In the last paragraph of the study the authors discuss implications for Antarctic ice stream dynamics. In particular, they mention observations of enhanced melting in ice stream margins, which provides significant motivation for the work presented in this study. I believe it may be useful as a takeaway for the reader to either expand on these observations and provide a clearer link between the work in this study and those observations or to suggest what these observations and the physical mechanism proposed in this study may mean for how we represent and model ice sheet dynamics.*

This is very valuable advice. We are eager to expand our manuscript, regarding the link between observations and our results and their meaning for modeling ice-sheet dynamics.

*Minor Comments:*

- *In the discussion of the results, I found myself losing track of the different perturbation experiments and some of the acronyms. It may be useful to have a table of the different experiments and the corresponding the melt rates.*

  We will add a table to give an overview of the different perturbation experiments and their characteristics, as suggested by the referee.

- *Line 22 on page 4: I wondered whether "efficiency of the melting" was a clear descriptor of Equation 1, rather than something like "sensitivity of the flux to melt rate".*

  We are glad to change the wording here according to the referee's suggestion.

- *Lines 7-12 on page 8: the comparison of melt rates in this study to melt rates estimated in ice shelves may be more useful in the "Setup and experimental design" section as a motivation for the choice of melt rates, as I found myself wondering how you chose the melt rates and whether they were physical*

  This is indeed a good idea and we will follow the referee's suggestion.

- *Does the width of the shear margin matter? If the shear margin is quite wide and thus velocities are going to zero slowly (i.e. if the flow law exponent is lower), would this dampen the effect of melting in the shear margin?*

  This is an interesting point raised by the referee. We will add a discussion of the influence of the shear-margin width.

*Citations*
*Alley, K.E., Scambos, T.A., Alley, R.B., Holschuh, N. (2019) Troughs developed in ice-stream shear margins precondition ice shelves for ocean-driven breakup. Science Advances, 5(10), doi: 10.1126/sciadv.aax2215*

*Suckale J, Platt JD, Perol T and Rice JR (2014) Deformation-induced melting in the margins of the West Antarctic ice streams. Journal of Geophysical Research: Earth Surface, 119(5), 1004–1025 (doi: 10.1002/2013JF003008)*

*Perol T and Rice JR (2015) Shear heating and weakening of the margins of West Antarctic ice streams. Geophysical Research Letters, 42(9), 3406–3413, ISSN 00948276 (doi: 10.1002/2015GL063638)*

*Haseloff M, Hewitt IJ and Katz RF (2019) Englacial Pore Water Localizes Shear in Temperate Ice Stream Mar- gins. Journal of Geophysical Research: Earth Surface, 124(11), 2521–2541, ISSN 2169-9003 (doi: 10.1029/ 2019JF005399)*

---

## Author Comment (AC3)

*The manuscript "Shear-margin melting causes stronger transient ice discharge than ice- stream melting according to idealized simulations" by Feldmann et al. investigates a relatively straightforward question: where does melting of ice shelves matter most? A lot of previous work has focused on the along-flow direction when addressing this question, while the authors focus on the across-stream direction. They apply localised melt either directly at the grounding line or in the shear margins. Maybe unsurprisingly they find that persistent melting matters most where the ice is slowest, which is in the shear margins of an ice shelf in their experiments.*

*The paper builds heavily on Reese et al. (2018) and is similar to Zhang et al (2020) and thus not overly novel in its approach. Nevertheless, I think it is worth pointing out that spatial variation in melting matters and to try to identify regions where melting is most influential.*

We are grateful for the willingness of Referee#3 to review our manuscript and appreciate their helpful suggestions and the constructive criticism. We agree with the referee that our approach is related to the studys cited by the referee, i.e., we investigate grounding-line flux sensitivity to basal ice-shelf-melt perturbations. However, we would like to note that our study is based on transient simulations. Thus our results provide insight on the time-dependent glacier response, which is not covered by the two mentioned studies. For instance, our simulations show a clear qualitative difference between the quasi-instantaneous response and the longer-term response.

*My main points of criticisms are:*

- *I think a more systematic investigation involving more locations would have greatly benefitted the paper and would have allowed a more systematic analysis of the role of distributed melt.*

  We agree with the referee that a more systematic investigation of melt regions would indeed be very interesting. However, we think that this would be beyond the scope of our study. Our work is intended to focus on the response difference to the two mentioned dynamically very different melt regions and the underlying physical mechanism which from our point of view deserves a study on its own. Therefore, and in the light of the considerable computational resources that have already been used for the conducted simulations, we would like to refrain from running further experiments. We would like to add a sentence to the conclusions section, stating that a systematic, transient analysis of the outlets of the Antarctic Ice Sheet would be an interesting next step.

- *The findings of the paper are really quite straightforward, and I don't see the need for 8 figures in the main text plus an additional 5 in the appendix to convey the results. Figures 1, 3, 4 and subsets of figures 5 and 6 would in my opinion suffice.*

  We see the referee's point here. At the same time, we think that omitting one or several figures would indeed mean a loss of information to the paper. For instance, Fig. 2, which shows the ice-velocity field, visualizes the regions of the ice stream's shear margins that are central to our study. Neglecting panels from Figs. 5 and 6 would neglect information regarding time evolution changes in the ice velocity and thickness. Fig. 7 covers the differences between the quasi-instantaneous and the longer-term response, which we deem very important. Fig. 8 summarises the flux sensitivity of all conducted experiments. Nevertheless, if the number of figures turns out to be a relevant point also for the editor, we would offer to shift Figs. 2, A1 and A3-A5 to the Supplement.

- *Ice stream shear margins are interesting for many authors because they are regions of enhanced warming with implications for ice flow and stability of ice shelves. I think this could be mentioned in the*

*text.*

We thank the referee for this valuable hint. We will include this information in our revised manuscript.

- *The paper title is a bit misleading -- being familiar with the large body of literature on ice stream shear margins, I didn't expect the paper to solely focus on isothermal ice shelf margins.*

We understand the point raised by the referee here. When choosing the manuscript title we decided to use the term "idealized simulations" in order to account for the simplified nature of our simulations and we would wish to keep it like that. However, if the referee/editor thinks that this is a crucial issue, we could imagine to change the title such that it states that the simulations are isothermal.

- *The paper is well-written, but somewhat selective (not to say negligent) in its discussion of existing literature. Relevant studies worth mentioning include (just to name a few)*
  - *Alley KE, Scambos TA, Alley RB, Holschuh N. Troughs developed in ice-stream shear margins precondition ice shelves for ocean-driven breakup. Science advances. 2019 Oct 1;5(10):eaax2215.*
  - *Alley KE, Scambos TA, Siegfried MR, Fricker HA. Impacts of warm water on Antarctic ice shelf stability through basal channel formation. Nature Geoscience. 2016 Apr;9(4):290-3.*
  - *Hunter P, Meyer C, Minchew B, Haseloff M, Rempel A. Thermal controls on ice stream shear margins. Journal of Glaciology. Cambridge University Press; 2021;67(263):435–49.*

We thank the referee for these important references that we missed. We will gladly add them to our manuscript and discuss them.

---

## Author Response (AR1)

**Response to the Editor**

**Dear Dr. Mantelli,**

We would like to thank you for the careful handling of the review process. We are grateful for the valuable comments and suggestions from you as well as from the referees as they really helped to improve our manuscript. Revising the manuscript, we took into account all the referees' suggestions (see our point-to-point answers below) but paid particular attention to the three main points raised by the Editor:

- 1) We now provide a more detailed presentation of the physics that underlie the comparatively strong flux response to shear-margin melting found in our simulations. To give these results also more visibility we restructured the text a bit, introducing the new subsection "Physics underlying the enhanced ice-flux sensitivity to shear-margin melting" (P5,L30 P7,L5). Though the Editor suggested to add such an individual section to the discussion we felt that it would fit better to the results section and hope the Editor agrees to it. This section now also covers results from a new set of simulations which we ran to test the influence of the melt-strip length *I* (in addition to the the width *w*) on the response (visualized in new Fig. S9). While addressing the referee's requests we added three further subheadings to the results/discussion sections, as we feel that they give the text more structure and a better orientation for the reader.
- 2) We have to admit that we really missed to reference and discuss some important shear-margin related studies and were glad to receive such valuable advice from the referees. We ensured to include and discuss all the literature suggested by the referees (observational and numerical) and also added further studies touching the shear margin topic. As a result, the revised version of the manuscript now also addresses the important topics of observed channelized melting beneath ice-shelf shear margins, the important general role of shear margins for ice-sheet/ice-shelf stability as well as relevant mechanisms like viscous heating and damage within the shear margins that are not accounted for by our simulations. To this end we introduced a new subsection "Further possible shear margin effects and model limitations" to the discussion/conclusions (P9,L11 P10,L5) and amended the Introduction (P2,L10-12).
- 3) To address the request of the Editor and Referee 3 for a more succinct presentation of our results we redesigned Figures 1, 3 and 4 7, strictly following the Editor's suggestions. Consequently, we also prepared a supplement which now includes original Figs. A1 A5 as well as the full versions of original Figs. 4 7. We also went through the text to provide a more precise referencing of the figures, as asked for by the Editor. In this context we also labeled figure panels with letters, where appropriate (Figs. 5, 6, S1 and S9)

Last but not least, we addressed all the Editor's valuable minor points in the commented pdf (see our responses in the same pdf). Please note that we slightly shortened the manuscript title (replacing "according to" by "in") to make it more concise.

Once again, we would like to acknowledge the work and the time the Editor and the referees put into the review process from which we think that it really enriched the manuscript. We hope that the Editor and the referees are content with our proposed revisions to the manuscript. Please find below the *referees' comments in italics* and our detailed response in blue.

Best wishes, J. Feldmann et al.

[revised manuscript text omitted]

---

## Editor Decision (ED1)

[revised manuscript text omitted]

---

## Author Response (AR2)

**Response to the Editor**

Dear Dr. Mantelli,

We are delighted to read that you accept our manuscript after minor revisions suggested by Referee #2. We gladly addressed the referee's points. Please find below the *referee's comments in italics* and our response in blue.

Best wishes,
J. Feldmann et al.

**Anonymous Referee #2**

*I want to thank the authors for their detailed responses and their revision of this paper. The revised sections in the introduction do a very good job of making the key takeaways of the paper clear. Further, the reworked section on the physics underlying the sensitivity to melting is very useful and does a good job of connecting with previous studies, the authors' results, and connections to fundamental glacier physics. Further, the reworked section on "Further possible shear-margin effects and model limitations" does a good job of connecting with shear margin studies and identifying what these results can and cannot say about ice dynamics, and I very much liked the descriptions of the effect of localizing shearing and the changing width of the shear margin. Overall, I believe that the authors have addressed my previous comments very well and have only a few minor suggestions to strengthen the readability of this paper.*

We thank the referee for taking the time to read the revised version of the manuscript and are delighted to read that the referee likes our changes.

*While I believe that the section on "further possible effects" does a good job of outlining the studies that look at effects of shear margin heating and melting, I believe this section would be stronger with a few statements on what effect not considering the temperature evolution in shear margins may have on the results presented in this study (and/or what results one might expect if you did account for thermal structures).*

We are grateful for this hint and extended Sec. 4.1, elaborating on what the inclusion of shear-margin heating in our simulations would mean for our results (P9, L14-22).

*There are a few sentences that were quite long and/or hard to parse (for example, page 7 line 5 and page 10 lines 8-9).*

Following the suggestion by the referee we split long sentences into two at several places in the manuscript (P7,L24; P7,L33; P8,L9; P9,L25) to improve readability.

*Finally, on page 12 line 7, I think the verb should be "involves" rather than "involve".*

Corrected.